

# Direct and legacy effects of plant-traits control litter decomposition in a deciduous oak forest in Mexico

Bruno Chávez-Vergara[1], Agustín Merino[2], Antonio González-Rodríguez[3], Ken Oyama[4] and Felipe García-Oliva[5]

[1] Instituto de Geología, Universidad Nacional Autónoma de México, Ciudad de Mexico, Mexico
[2] Escuela Politécnica Superior, Universidad de Santiago de Compostela, Lugo, Galicia, Spain
[3] Instituto de Investigaciones en Ecosistemas y Sustentabilidad, Universidad Nacional Autónoma de México, Morelia, Michoacán, Mexico
[4] Escuela Nacional de Estudios Superiores Unidad Morelia, Universidad Nacional Autónoma de México, Morelia, Michoacán, Mexico
[5] Instituto de Investigaciones en Ecosistemas y Sustentabilidad, Universidad Nacional Autónoma de México, Morelia, Michoacán, Mexico

Corresponding author
Felipe García-Oliva,
fgarcia@cieco.unam.mx

## ABSTRACT

**Background**. Litter decomposition is a key process in the functioning of forest ecosystems, because it strongly controls nutrient recycling and soil fertility maintenance. The interaction between the litter chemical composition and the metabolism of the soil microbial community has been described as the main factor of the decomposition process based on three hypotheses: substrate-matrix interaction (SMI), functional breadth (FB) and home-field advantage (HFA). The objective of the present study was to evaluate the effect of leaf litter quality (as a direct plant effect, SMI hypothesis), the metabolic capacity of the microbial community (as a legacy effect, FB hypothesis), and the coupling between the litter quality and microbial activity (HFA hypothesis) on the litter decomposition of two contiguous deciduous oak species at a local scale.

**Methods**. To accomplish this objective, we performed a litterbag experiment in the field for 270 days to evaluate mass loss, leaf litter quality and microbial activity in a complete factorial design for litter quality and species site.

**Results**. The litter of *Quercus deserticola* had higher rate of decomposition independently of the site, while the site of *Quercus castanea* promoted a higher rate of decomposition independently of the litter quality, explained by the specialization of the soil microbial community in the use of recalcitrant organic compounds. The Home-Field Advantage Index was reduced with the decomposition date (22% and 4% for 30 and 270 days, respectively).

**Discussion**. We observed that the importance of the coupling of litter quality and microbial activity depends on decomposition stage. At the early decomposition stage, the home-advantage hypothesis explained the mass loss of litter; however, in the advanced decomposition stage, the litter quality and the metabolic capacity of the microbial community can be the key drivers.

## INTRODUCTION

Litter decomposition is a key process in the functioning of forest ecosystems, because it strongly controls nutrient recycling and soil fertility maintenance (*Austin et al., 2014*). At the local scale, the decomposition rate is strongly affected by litter traits and microbial activity (*Freschet, Aerts & Cornelissen, 2012*). The litter traits that promote the decomposition are related with physical features such as the rate of water uptake in litter (*Makkonen et al., 2013*). Additionally, some chemical characteristics of litter can promote its decomposition such as: (a) a low C: N ratio (*Agren et al., 2013*; *Aponte, García & Marañón, 2013*; *Bonanomi et al., 2013*; *Osono, J-i & Hirose, 2013*), (b) a high concentration of soluble organic forms (*Berg, 2014*), and (c) a low proportion of lignin or phenolic compounds (*Almendros et al., 2000*; *Prescott, 2010*; *Ono et al., 2011*), as well as changes in the proportions of lignin subunits (*Chavez-Vergara et al., 2014*; *Talbot et al., 2012*). The study of the effects of leaf litter traits on decomposition has been reported before by several authors (i.e., *Grime & Anderson, 1986*; *Baas, 1989*; *Grime et al., 1996*); and more recently, these traits are called ''after life'' traits, because they are products of the metabolism of living plant species, and they can regulate ecological processes, such as litter decomposition (*Genung, Bailey & Schweitzer, 2013*).

The decomposition rate of organic compounds is also associated with the composition of the soil microbial community and its metabolism (*Austin et al., 2014*; *Freschet, Aerts & Cornelissen, 2012*). For example, the presence of actinomycetes (*Snajdr et al., 2011*) and basidiomycetes species (*Osono & Takeda, 2002*; *Snajdr et al., 2011*) favors the degradation of recalcitrant compounds (i.e., lignin, polyphenols, aliphatics), because these microbial taxa are capable of producing exoenzymes which can cleave these organic molecules (*Allison, Chacon & German, 2014*). Consequently, the inhibitory effect on litter decomposition of a high proportion of recalcitrant molecules can be reduced by the activity of specialized microbial species (*Cleveland et al., 2004*; *Strickland et al., 2009*; *Snajdr et al., 2011*; *Chávez-Vergara et al., 2016*). Therefore, the metabolic capacity of the microbial community can be considered as a ''legacy'' effect over litter decomposition (*Wurst, Ohgushi & Allen, 2015*). These authors defined legacy effect as ''a specific case of long-term effects that persist after the biotic interaction that caused the effects ceases''. Recent studies have shown that the interaction between the chemical composition of the plant residues and the metabolic capacity of the microbial community is the most important factor in the regulation of the litter decomposition rate (*Austin et al., 2004*; *Ayres et al., 2009*; *Fanin, Fromin & Bertrand, 2016*; *Garcia-Palacios et al., 2016*; *Hicks Pries et al., 2017*). This interaction involves the functional traits of plant species (i.e., chemical characteristics of plant residues) and the activity of the microbial community of the forest floor (i.e., production of exoenzymes) (*Ayres et al., 2009*; *Austin et al., 2014*; *Pearse et al., 2014*; *Fanin, Fromin & Bertrand, 2016*).

The coupling between litter chemical composition and metabolism of the microbial community of the forest floor has been described by the following hypotheses: (A) home-field advantage (HFA), which states that the litter will be more easily decomposed by the microbial community in the same site where it was produced (*Ayres et al., 2009*; *Austin et al., 2014*); (B) substrate-matrix interaction (SMI), in which exogenous litter can

be decomposed at the same rate as endogenous litter if both have a similar chemical composition (*Freschet, Aerts & Cornelissen, 2012*) and, more recently, (C) the functional breadth hypothesis (FB), according to which microbial communities that have been exposed to substrates with low chemical quality have developed mechanisms for the use of substrates with different chemical quality; in other words, are functionally more diverse, and capable of using a wide-range of substrates (*Fanin, Fromin & Bertrand, 2016*). Therefore, the objective of the present study was to evaluate the effect of leaf litter quality (as a direct plant effect, SMI hypothesis), the metabolic capacity of the microbial community (as a legacy effect, FB hypothesis), and the coupling between leaf litter quality and microbial activity (HFA hypothesis), on the litter decomposition of two species of deciduous oaks, by using a controlled field experiment of litter decomposition.

In previous studies, we found that *Q. deserticola* promoted higher nutrient availability than *Q. castanea*, because the former oak species produced leaf litter with higher chemical quality, therefore favoring microbial activity and litter chemical transformation (*Chavez-Vergara et al., 2014*; *Chávez-Vergara et al., 2015*). However, the microbial community in the litter of *Q. castanea* is dominated by microbial species specialized in the use of recalcitrant compounds, increasing the efficiency in the use of resources (*Chávez-Vergara et al., 2016*). Therefore, the main hypothesis of the present study is that the litter decomposition is regulated by the direct effect of the chemical composition of the plant residues, and the legacy effect on the specialization of the microbial community. Consequently, the site dominated by *Q. castanea* should have a higher potential of litter decomposition, but the *Q. deserticola* litterfall should be easier to decompose. Our study is the first report testing hypotheses on litter decomposition in species of the same genus, while most of the studies have been performed on taxonomically and functionally very distant plant species (i.e., *Freschet, Aerts & Cornelissen, 2012*; *Pearse et al., 2014*; *Fanin, Fromin & Bertrand, 2016*).

To test our hypothesis, we performed a factorial field experiment of decomposition bags during 30 and 270 days with litterfall of *Q. castanea* (low quality), *Q. deserticola* (high quality) and a mix of both oak species litterfall (cumulative quality) in three sites based on the microbial activity specialization: fast degradation of recalcitrant compounds (under *Q. castanea)*, slow degradation of recalcitrant compounds (under *Q. deserticola*) and an intermediate degradation, which represents a wider spectrum of resource utilization for the microbial community (under both *Quercus* species in interaction).

## MATERIALS AND METHODS
### Study site
This study was conducted within the Cuitzeo basin in El Remolino hill (19°37′01″N, 101°20′07″W; 11 km south of Morelia city, Michoacán, Mexico). The study site is an oak forest fragment (>12 ha) with low disturbance (about 80 years without wood extraction for charcoal production according to nearby inhabitants) and two dominant native oak species: *Q. castanea* Née (section *Lobatae*) and *Q. deserticola* Trel. (section *Quercus*). The characteristics of this site and the species can be found in more detail in previous studies

(*Chavez-Vergara et al., 2014*; *Chávez-Vergara et al., 2015*). Briefly, the predominant soil type is a chromic Luvisol developed over Quaternary basalts. The climate in the area is temperate subhumid, with annual mean temperature of 17.6 °C and annual mean precipitation of 805 mm concentrated in the summer months. In 2014, the annual rainfall was 850 mm and the average temperature was 16.6 °C. For the present study, three parallel plots of 30 × 150 m were established perpendicular to the main slope, where one of the studied species dominated in either of the two lateral plots; both species were mixed in the central plot. Therefore, three species conditions were present: isolated *Q. castanea* (Qc), isolated *Q. deserticola* (Qd) and mixed *Quercus* species (Qx).

## Litterfall collection

A circular trap of 0.5 m² was placed under each of five trees per species condition: isolated *Q. castanea*, mixed species and isolated *Q. deserticola* (15 traps in total) to collect litterfall every month from December 2012 to May 2014. The fresh litter samples were weighed, and an aliquot was dried to constant weight at 70 °C for 72 h to determine water content, which was then used to calculate the dry mass of each sample. The fresh aliquot was stored at 4 °C in darkness prior to laboratory analysis ($n = 5$ for each species). The monthly dried subsamples from the two sampling years were mixed for the litterbag experiment.

## Litterbags experiment
### Field experiment
Brown color polyester mesh (1 mm) bags (10 × 10 cm) were used for the field decomposition experiment. The mesh size of 1 mm was chosen because it avoids losing small leaf litter debris but allows the activities of the aerobic microbial community and meso- and micro-fauna (*Nguye Tu et al., 2011*), which play an important role in the initial fragmentation of litter (*Gessner et al., 2010*). These bags were filled with 11 g of oven-dried litterfall with the following arrangement: 30 bags with *Q. castanea* litterfall (QcL), 30 bags with *Q. deserticola* litterfall (QdL) and 30 bags with a mixture of both species litterfall (QxL) in the same proportion (5.5 g of *Q. castanea* and 5.5 g of *Q. deserticola* litterfall). In June 2014, two litterbags of each litterfall type (QcL, QdL and QxL) were randomly located above the litter and around the stem of each of the five selected trees, distributed along the main slope, in each species condition plot (hereafter plots are referred to as sites): *Q. castanea* site (QcS), *Q. deserticola* site (QdS) and the species mixture site (QxS). Therefore, the field design is a complete factorial 3 × 3 (site and litterfall conditions). One litterbag for each treatment per tree was harvested at 30 and 270 days after the bags were placed (five bags for each treatment). The comparison of the two dates allows us to determine the decomposition effect on early and late decomposition stages, where the labile and recalcitrant molecules proportion changes over time. The means (± standard deviation) of DBH for trees in each condition were Qc: 52.9 ± 11.7 cm, Qx: 48.9 ± 4.9 cm and Qd: 63.9 ± 11.4 cm.

In the collection dates, the content of each bag was carefully removed, fresh field weighed, and subsequently divided into two subsamples. The first one was stored in hermetic bags in the dark at 4 °C until laboratory analysis. The second subsample was dried to constant

weight at 70 °C for 72 h to calculate the water content. Then, the sample was milled in a ball mill at 350 RPM for three min and stored in sealed bags until chemical analysis.

### Remaining mass and decomposition rate

Initial and remaining samples were combusted in a muffle furnace at 650 °C to determine inorganic particles to correct data on an ash-free basis. The litter decomposition rate was calculated using the simple exponential model: $MR = Mi\ e^{-kt}$; where MR is the percentage of the remaining mass at 270 days, Mi is the initial mass percentage, $k$ is the decomposition rate and t is the decomposition time in field conditions (*Olsen, 1963*).

## Laboratory analysis

The nutrients and enzymatic activity analyses were done at Instituto de Investigaciones en Ecosistemas y Sustentabilidad, UNAM, Mexico, while the $^{13}$C Nuclear Magnetic Resonance and DSC analyzes were done at Universidad de Santiago de Compostela, Spain.

### $^{13}$C Nuclear Magnetic Resonance ($^{13}$C NMR) spectroscopy

The $^{13}$C Nuclear Magnetic Resonance spectroscopy is a non-destructive analysis that improves the identification of the molecular composition from organic residues; it is useful tool for determination of the molecular composition of litterfall and decomposed litter. To characterize the chemical composition of litterfall and decomposed litter, the analysis of Cross Polarized Magic-Angle Spin $^{13}$C NMR in solid state was performed in samples previous to the field experiment (initial) and in samples at the end of the field experiment (remaining). The $^{13}$C NMR data were obtained at 298 K in a Varian Inova-750 17.6 T (operated at 750 MHz frequency proton), under the conditions described in *Chavez-Vergara et al. (2014)*. The spectrogram obtained was processed with the program MestreNova V. 6 (Mestrelab Research Inc.).

For integration, the spectrogram was divided into four major regions representing different chemical environments of $^{13}$C nucleus according to position of relaxation signal in parts per million of chemical shift (ppm): C Alkyl (0–45 ppm), O-alkyl C (45–110 ppm), aromatic C (110–160 ppm) carbonyl and C (160–220 ppm). For more detailed analysis, spectra were divided according to *Leifeld & Kögel-Knabner (2005)* as: (I) 10–45 ppm C alkyl: methyl groups, methylene groups on rings and aromatic chains. (II) 45–110 ppm C O-alkyl: methoxy groups and C6 in some polysaccharides (45–60 ppm); C2–C5 hexoses C of some amino acids, aliphatic alcohols and fractions of lignin structure (60–90 ppm); Carbohydrate anomeric C, C2–C6 syringyl unit of lignin (90–110). III) 110–160 ppm aromatic C: and CC and CH carbon C2 guaiacil, C6 lignin (110–140 ppm, aryl C); COR aromatic or CNR (140–160 ppm, phenolic C) groups. IV) carboxyl 160–220 ppm C: carboxyl C, C carbonyl and C amide.

We also examined indexes associated with the decomposability of organic matter based on integrated specific regions: alkyl: O-alkyl ratio (A: OA), O-alkyl: aromatic ratio (OA: Ar), aromaticity (Ai), hydrophobicity (HB: HI) and characterization of lignin relations based on subunits specific regions such as syringyl (S), guaiacyl (G) and p-hydroxyphenyl (H) as lignin relations S:G, S:H and G:H (*Almendros et al., 2000*; *Spaccini et al., 2006*; *Talbot et al., 2012*; *Bonanomi et al., 2013*; *Chavez-Vergara et al., 2014*).

### Differential Scanning Calorimetry (DSC) and Thermogravimetry (TGA)

The Differential Scanning Calorimetry (DSC) and Thermogravimetry (TG) is a thermal analysis suitable for determination of organic matter stability (*Angehrn-Bettinazzi, Lüscher & Hertz, 1988*). This method quantifies the energy release during different combustion temperatures of samples, like the energy required for biological oxidation of organic molecules (*Rovira et al., 2008*). Therefore, the thermograms can quantify the proportion of labile, recalcitrant and extra-recalcitrant compounds in the organic samples (*Barros, Salgado & Feijóo, 2007*). The characterization of thermal properties of litterfall and decomposed litter was done by differential scanning calorimetry and thermogravimetric analysis (DSC-TGA, Mettler-Toledo International Inc.). The analysis was performed with 4 mg of powdered oven-dried sample placed in an aluminum pan in an atmosphere of dry air (flow rate, 50 ml min$^1$) and the scan rate was 10 °C min$^{-1}$. The temperature range used was 50 to 600 °C. An indium sample (melting point: 156.6 °C) was used to calibrate the calorimeter. All samples were analyzed in triplicate.

The combustion heat release (Q, J g$^{-1}$) was determined by integrating the DSC curves (W g$^{-1}$) on the exothermic region (150–600 °C). Data recorded at temperatures <150 °C were discarded because they are associated with the loss of mass and energy release during moisture loss. The Q value was divided by the mass loss in each measurement (Q′, J mg$^{-1}$ MO, *Rovira et al., 2008*).

Areas under the DSC curve were divided in three groups, representing different degrees of resistance to thermal oxidation (*Dell' Abate et al., 2002*; *Fernández et al., 2012*): labile organic matter, comprising of carbohydrates and other aliphatic compounds (200–375 °C); recalcitrant organic matter such as lignin and/or polyphenols (375–475 °C); and extra-recalcitrant organic matter, such as polycondensed aromatic forms (475–550 °C). The heat release by combustion in each region was designated as $Q_1$, $Q_2$ and $Q_3$, respectively. Also, the temperature at which the maximum heat flow was detected during the combustion of organic matter ($T_1$, $T_2$ and $T_3$) and the temperature at which 50% of the total energy was released were recorded ($T_{50}$DSC).

### Nutrient analysis

All forms of C were determined on a total carbon analyzer (UIC model CM5012, Chicago, IL, USA) by dry combustion and coulometric detection (*Huffman, 1977*), while forms of N were determined colorimetrically by the semi-Kjeldahl (*Bremmer, 1996*) method and the forms of P by molybdate colorimetric method after reduction with ascorbic acid (*Murphy & Riley, 1962*) in a Bran-Luebbe autoanalyzer (Autoanalyzer 3 Norderstedt, Germany) after acid digestion. The litterfall chemical analyses were performed from material collected from the respective traps (isolated *Q. castanea*, mixed species, and isolated *Q. deserticola*).

Soluble organic forms of C, N and P were extracted from 2 g of fresh material in deionized water, after stirring for 1 h and filtered through a Whatman# 42 filter and on a vacuum system through a 0.45 μm nitrocellulose membrane. The dissolved organic carbon (DOC) was determined by combustion coulometric detection (*Huffman, 1977*). Dissolved organic nitrogen (DON) and dissolved organic phosphorus (POD) were determined after acid

digestion. The DON was calculated as the difference between the acid digested nitrogen and soluble $NH_4^+$, as well as POD (acid digested P minus soluble inorganic P; *Joergensen & Mueller, 1996*).

Microbial biomass carbon (Cmic) and nitrogen (Nmic) were determined by direct extraction using chloroform fumigation (*Brookes, Powlson & Jenkinson, 1984*; *Vance, Brookes & Jenkinson, 1987*) from fresh samples. Two subsamples (2 g) were incubated at 30 °C for 24 h; one of the subsamples was maintained in chloroform atmosphere during incubation. Both samples were extracted in 0.5 M $K_2SO_4$ and percolated through # 42 Whatman filter paper and analyzed as DOC (as mentioned above) and removable N. The extractable N was quantified as total N after acid digestion (*Brookes, Powlson & Jenkinson, 1984*).

The microbial biomass P (Pmic) was determined by the fumigation-extraction method (*Brookes, Powlson & Jenkinson, 1982*) in fresh samples. A subsample of 0.5 g was fumigated for 24 h in a chloroform atmosphere and extracted with 30 mL of 0.5 M $NaHCO_3$ pH 8.5 for 30 min (*Van Meeteren, Tietema & Westerveld, 2007*). Extracts (fumigated and no-fumigated) were digested in a solution of sulfuric acid and persulfate ammonium according to Hedley sequential P fractionation (*Tiessen & Moir, 2008*) and quantified as orthophosphate, as described above. Cmic concentration, Nmic and Pmic were calculated from the difference between the fumigated and non-fumigated samples, then the concentration was corrected by applying the following factors: KeC 0.45 (*Sparling et al., 1990*), KeN 0.54 (*Brookes, Powlson & Jenkinson, 1984*; *Joergensen & Mueller, 1996*) and KeP 0.40, respectively (*Brookes, Powlson & Jenkinson, 1982*).

### Enzyme activity

As a measurement of microbial activity related with the use of organic molecules, the enzymatic activities of $\beta$-1,4-glucosidase (BG), cellobiohydrolase (CBH), $\beta$-N-acetyl-glucosaminidase (NAG), polyphenol oxidase (POX) and dehydrogenase (DHG) were determined in fresh material of each treatment for all samples collected. The determination of hydrolases (BG, CBH and NAG) was performed according to *Chavez-Vergara et al. (2014)* by colorimetric measurement of p-nitrophenol (pNP) in a spectrophotometer (Evolution 201, Thermo Scientific Inc., Waltham, MA, USA) at 420 nm liberated from specific substrates during incubation (2 h in oscillatory shaking at 30 °C) and reported in g-mol pNP litter$^{-1}$ h$^{-1}$.

POX activity was determined through oxidation of $2, 2'$-Azinobis [3-ethylbenzothiazoline-6-sulfonic acid]-diammonium salt (ATBS). One aliquot of the same extraction used for activity of hydrolases (*Chavez-Vergara et al., 2014*) was used, volume and time of preparation were the same as for hydrolases but in this case the result of the centrifugation was measured directly (without addition of NaOH and deionized water) on the same spectrophotometer described above by colorimetry at 460 nm and the result was reported as mol tyrosine g-litter$^{-1}$ h$^{-1}$, in this case the calibration curve is derived from tyrosine (tyr).

The dehydrogenase activity (DHG) was determined according to a modification of the method described by *Alef (1995)*, which is based on the reduction of chloride of 2, 3, 5-triphenyltetrazolium (CTT) for the formation of triphenyltetrazolium formazan (TFF)

in incubation at 30 °C for 24 h. To perform the assay 0.5 g of milled fresh material was weighed and placed in a conical tube with a capacity of 15 mL (the tube was covered with foil to keep out light) to which 1 mL of a solution CTT 1% in Tris buffer pH 7.6 was added. Blank samples were prepared only with the solution of CTT. Subsequently, all samples were placed and horizontally fixed in an incubator chamber; the incubation was for 24 h at 30 °C and 180 rpm. After incubation, 10 mL of acetone was added; the tube was stirred vigorously and allowed to react for 2 h in the dark at room temperature. Following, the supernatant was filtered through Whatman # 42 paper and measured at 546 nm by colorimetry. The measure of each sample was subtracted from the average value of blanks and adjusted by the equation described in *Alef (1995)*. The results were expressed in g TPF $g^{-1}d^{-1}$.

The efficiency of enzymatic activity according to the concentration of a nutrient immobilized in the microbial biomass was calculated as specific enzyme activity (SEA) according to the following equation (*Chavez-Vergara et al., 2014*):

$$SEA = A/Bmic \tag{1}$$

where SEA is expressed in mol of pNP or mol of tyr released per milligram of nutrient in the microbial biomass per hour (mol $mg^{-1}$Bmic $h^{-1}$); A is the activity of any of the specific enzymes (BG, CBH, NAG, POX and DHG), and Bmic is the concentration of Cmic or Nmic in mg $g^{-1}$. The association of enzymes with nutrients in the microbial biomass is as follows: BG, CBH, POX and DHG are associated with Cmic, and NAG with Nmic.

## Statistical analysis

The initial nutrient concentration in the three litterfall conditions had five replicates, while the DSC and the $^{13}$C NMR parameters had only one composite sample for each *Quercus* species, as well as the DSC and $^{13}$C NMR parameters for the samples at 270 decomposition days. In contrast, the DSC parameter of the 30 decomposition days had a composite sample for each litter condition (litter origin and litter site, in total nine samples). Therefore, the nutrient concentration of litterfall and DSC parameters according to litter quality and site effects at 30 days of decomposition were analyzed using one-way analyses of variance (ANOVA), while the remaining mass, total nutrient concentration and dissolved nutrients were analyzed with repeated measures ANOVA (RMANOVA), in which the between factor was litter species condition (QcL, QxL and QdL) and the within factors where sampling date (litterfall, and litter after 30 and 270 days of field decomposition, $n = 5$) and the interaction between litter condition and decomposition time. The data of nutrients, enzymatic activities and specific enzymatic activities at 30 and 270 days of experiment were analyzed by a factorial ANOVA to test the effect of litter condition (QcL, QdL and QxL) and site (QcS, QdS and QxS) in early and late decomposition. In this case, all samples associated with each treatment in the factorial design ($n = 15$) were used. Afterwards, statistically significant differences in one-way, RMANOVA and factorial ANOVA were analyzed with the Tukey HSD post-hoc test.

To identify the relationship between the chemical quality of the litter and microbial metabolism variables in the remaining mass at 30 days of decomposition, we performed

backward stepwise multiple regression analyses. As chemical quality variables, the C:N and C:P ratios, dissolved forms of C, N and P were used, microbial activity variables, the specific enzymatic activities, were used for the multiple regression model. We performed a correlation model using remaining mass and thermal parameters from the DSC analysis as indicators of litter changes during early decomposition process. All analyses were made in the statistical package Statistica 7.0 (StatSoft, USA).

The Home-Field Advantage Index (HFAI) proposed by *Ayres et al. (2009)* was calculated as the percentage of mass loss of each litterfall condition (QcL, QXL, QdL) at the site where it was produced relative to the mass loss at all sites (QcS, QXS, QdS):

$$A_{RMLa} = \frac{Aa}{Aa + Ba + Ca} \times 100 - 100 \tag{2}$$

where, $A_{RMLa}$ represents the relative mass loss of litter from condition $A$ at site $a$, and $Aa, Ba$ and $Ca$ represent the mass loss of litter from conditions $A, B$ and $C$ decomposing at site a, respectively. The measures of relative mass loss for each condition and site combination were used to calculate HFAI.

$$HFAI = \left[ \frac{A_{RMLa} + B_{RMLb} + C_{RMLc}}{3} \Big/ \frac{A_{MRLb} + A_{RMLc} + B_{RMLa} + B_{RMLc} + C_{RMLa} + C_{RMLb}}{6} \right)$$
$$100 - 100. \tag{3}$$

# RESULTS

## Mass loss and residence time

Figure 1A shows that *Q. deserticola* had higher decomposition rate at all harvest dates in 270 days of field decomposition experiment, but not at 30 days (Fig. 1A), and the estimated residence time of litter for the three conditions was 1.2, 1.3 and 1.9 years for *Q. deserticola* (Qd), mixed species (Qx) and *Q. castanea* (Qc), respectively. We observed that at 30 days of decomposition, the slope of mass loss was higher in Qd and Qx than in Qc (Fig. 1A). For this reason, we analyzed the mass loss at 30 and 270 days with a factorial ANOVA to identify the litter and site effect over mass loss. We observed that the mass loss after 30 and 270 days showed differences for the two main factors analyzed in opposed ways: Qd litterfall (QdL) promotes higher mass loss (Fig. 1B), while in the Qd site (QdS) lower mass loss was observed (Fig. 1C). The Home-Field Advantage index was 22% and 4% for 30 and 270 days, respectively.

## Chemical composition of litterfall and decomposed litter
### $^{13}C$ CP MASS NMR characterization of litterfall samples

Figure 2 shows the spectra and Table 1 the integration of regions of $^{13}$C CPMAS NMR of litterfall and decomposed litter at 270 days in both species. In the original litterfall, the most prominent compounds were O-Alkyl, Aryl C and Alkyl C. Particularly, we observed that the region between 160 and 220 ppm, assigned to carboxyl/amide and carbonyl C groups, is dominated by a peak at ca. 173 ppm (Fig. 2) with similar relative C distribution for both species (Table 1). In the aromatic and phenolic region (160–110 ppm), the litterfall produced by both species showed well defined peaks at ca. 153 and 145 ppm, which revealed

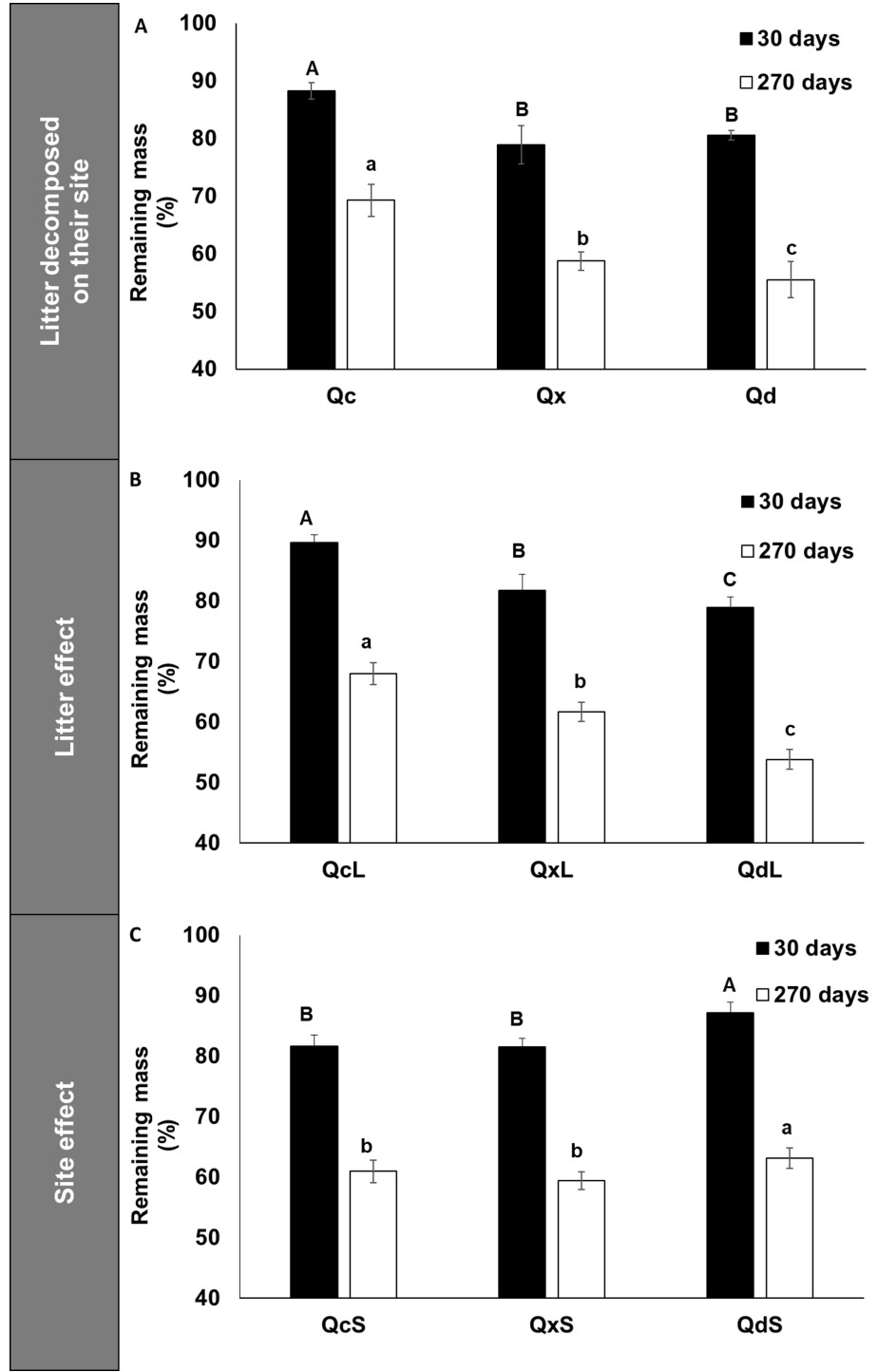

**Figure 1 Remaining mass in litterbags after 30 and 270 decomposition days.** (A) Remaining mass in litterbags after 30 and 270 days of decomposition in the field for each litter condition decomposed in the same site of production, (B) Effect of litter quality over remaining mass at 30 days and 270 days, and (C) Effect of site over remaining mass at 30 and 270 days. Different capital letters indicate differences among conditions at 30 days and lowercase letters indicate differences among conditions at 270 days.

**Table 1** Chemical characterization of litterfall and decomposed material in litterbags at 270 days by $^{13}$C CPMAS NMR.

| Chemical shift (ppm) | Q. castanea | | Q. deserticola | |
|---|---|---|---|---|
| | Litterfall | 270 days | Litterfall | 270 days |
| **Principal regions** | | | | |
| • Alkyl C (0–45 ppm) | 11 | 11 | 14 | 18 |
| • O-Alky C (45–110 ppm) | 66 | 64 | 63 | 62 |
| • Aryl C (110–160 ppm) | 18 | 17 | 17 | 15 |
| • Carboxyl C (160–220 ppm) | 6 | 7 | 6 | 6 |
| **Ratios and indexes** | | | | |
| • Alkyl C:O-Alkyl C | 0.17 | 0.17 | 0.22 | 0.29 |
| • O-Alkyl C:Aromatic C | 3.66 | 3.74 | eb3.71 | 4.13 |
| • Hidrophobicity (HB:HI) | 0.40 | 0.39 | 0.45 | 0.48 |
| • Aromaticity | 0.23 | 0.22 | 0.22 | 0.18 |
| • S:G | 1.35 | 1.75 | 1.00 | 1.50 |
| • S:H | 2.30 | 1.75 | 1.50 | 1.50 |
| • G:H | 1.18 | 1.00 | 1.45 | 1.00 |

**Notes.**

HB, hydrophobic compounds; HI, hydrophilic compounds; S, syringyl; G, guaiacyl; H, $p$-hidroxypheny.

the presence of C3, C5 syringyl lignin and tannins, respectively. An incipient peak at ca. 131 in *Q. deserticola* and well-defined in *Q. castanea* may be related to unsubstituted and C-substituted phenyl carbon of lignin monomers of syringyl units (*De Marco et al., 2012*) affecting the S:G ratio between species (Table 1). In the O-alkyl region (45–110 ppm), the most prominent signal was at ca. 73 ppm and it was particularly associated to the simultaneous resonance of C-2, C-3 and C-5 of pyranose rings in cellulose and hemi-cellulose; and a second prominent peak was at ca. 105 ppm, traditionally associated to crystalline cellulose, and it can be associated to non-protonated carbon arising from tannins (*Almendros et al., 2000*).

A shoulder at 56 ppm, attributable to N-methoxyl C compounds in lignin, was relatively well resolved in litter samples of both species. Meanwhile, the C-6 position in the pyranose ring of cellulose produced the peak or shoulder at ca. 64 ppm better resolved in *Q. castanea*. In a similar way, that shoulder at 84 ppm may correspond to C-4 in cellulose (*Almendros et al., 2000*). The alkyl region (46–0 ppm) showed two well-defined peaks at 30 and 21 ppm. The peak at ca. 30 ppm is related to polyethylene carbons in lipids and lipid polymers such as cutine or suberine and the peak at ca. 21 ppm is frequently attributed to acetate groups in hemicellulose and/or short chain aliphatic structures. The most noticeable changes after 270 days of decomposition occur in the reduction of peak intensity in ca. 145 ppm associated to tannins and the increment of peaks at ca. 56 ppm (N-methoxyl) and two peaks in the alkyl C region (Fig. 2). These changes were most intense in the *Q. deserticola* litter.

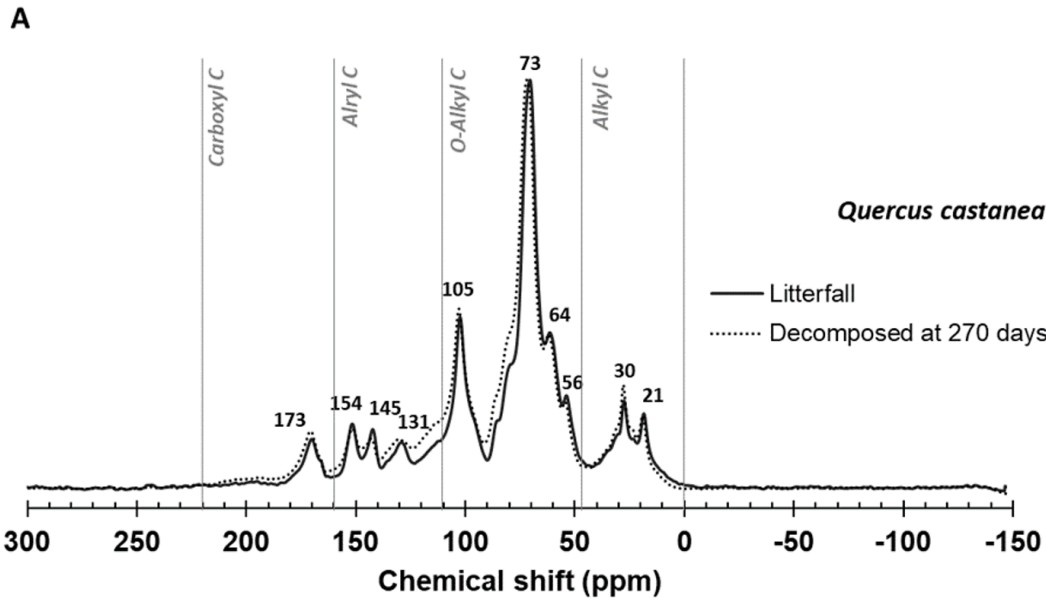

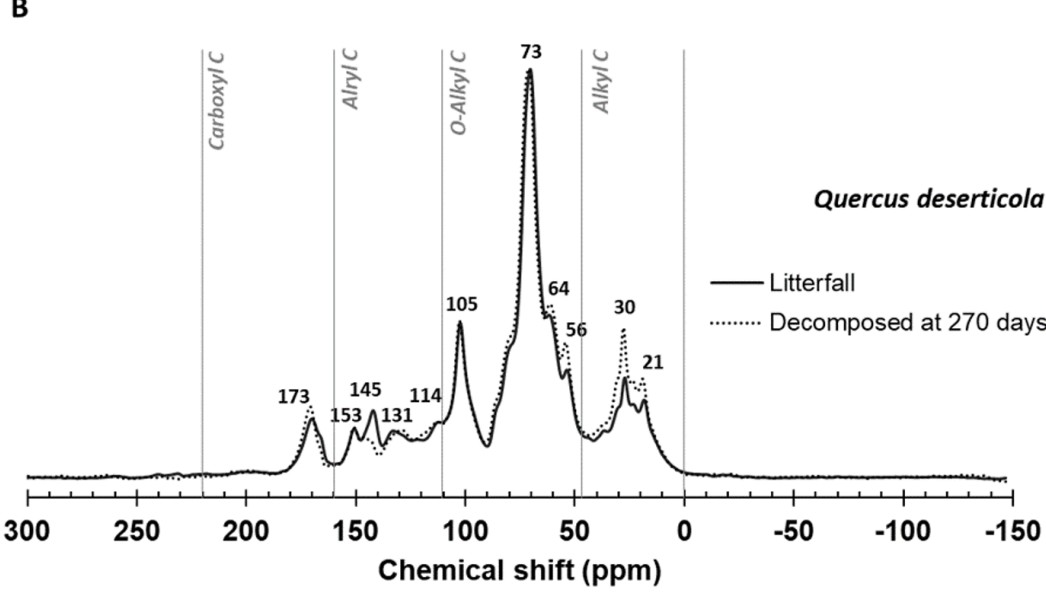

**Figure 2** **¹³C CPMAS NMR spectrograms.** Litter decomposing on their site: (A) *Quercus castanea* litterfall (solid line) and decomposed litter (dotted line) and (B) *Quercus deserticola* litterfall (solid line) and decomposed litter (dotted line).

### Thermal characteristics of litterfall samples by differential scanning calorimetry (DSC)

The litterfall thermograms of both species showed a bimodal shape (Figs. 3A and 3B). The first peaks were situated at 347 °C and 352 °C, and the second prominent peaks were at 449 °C and 461 °C for *Q. castanea* and *Q. deserticola*, respectively (Table 2). Additionally, the Qc litterfall had a second peak at 424 °C in the Q2 region. However,

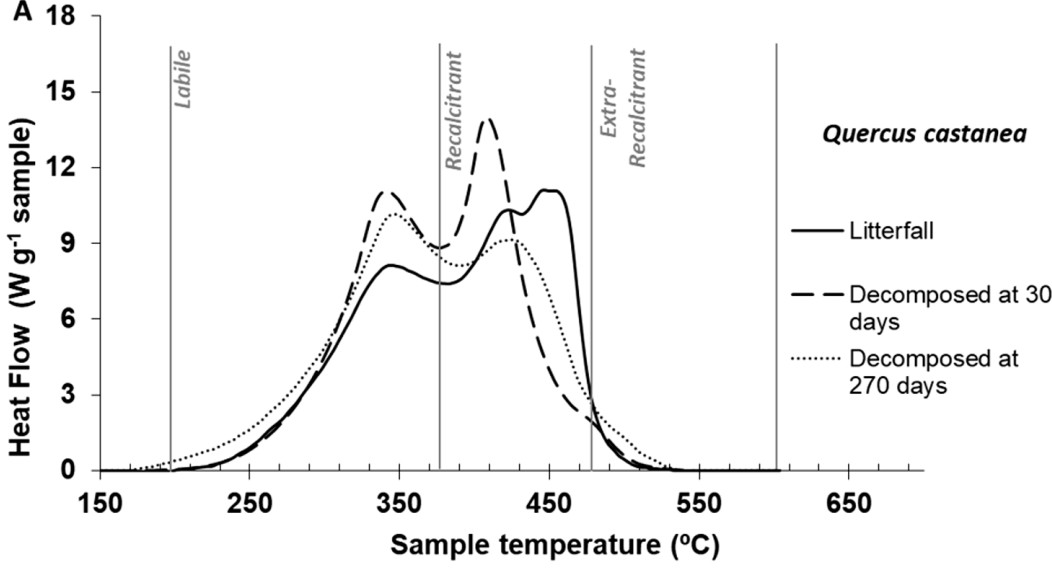

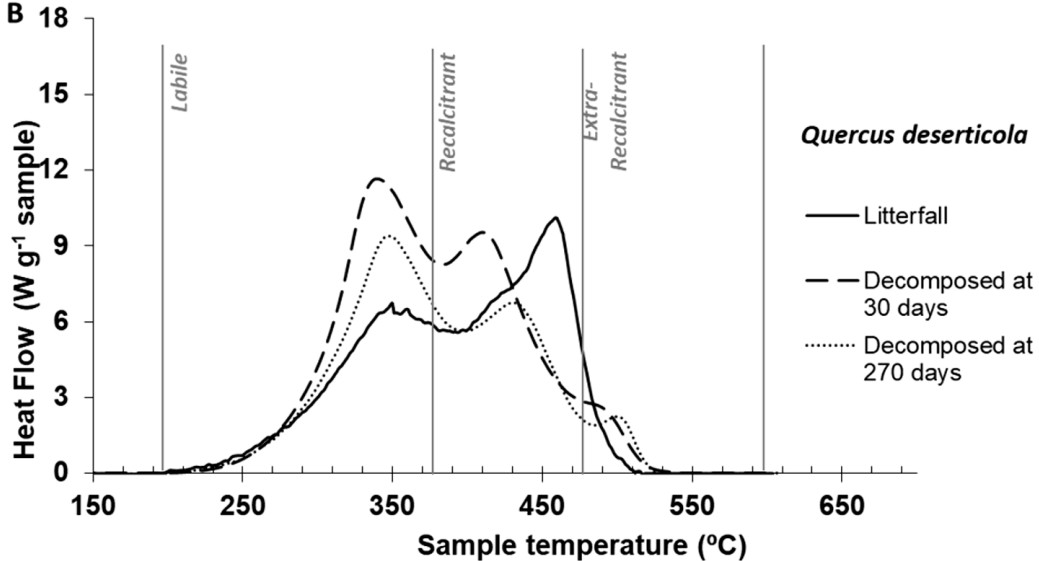

**Figure 3  DSC thermogams.** Litter decomposing on their site: (A) *Quercus castanea* and (B) *Quercus deserticola* litterfall (solid line), decomposed at 30 days (broad dotted lines) and decomposed at 270 days (fine dotted lines).

the energy released from the recalcitrant region (Q2) was higher in the *Q. castanea* than in the *Q. deserticola* litterfall (Table 2). After 30 days of decomposition, the percentage of recalcitrant compounds (Q2) was higher in the Qc litter than in the Qd litter, while the Qd litter had the highest percentage of extra-recalcitrant compounds (Q3; Table 2; Figs. 3A and 3B). In the 270 days decomposed litter, the shape of the thermogram of *Q. castanea* remained as bimodal, while for *Q. deserticola* the third peak at the extra-recalcitrant region is well-defined and displaced to a higher temperature (503 °C; Fig. 3). The decomposed

**Table 2** Thermal characteristics of litterfall and decomposed material in litterbags at 30 and 270 days analyzed by DSC-TG.

|  | Q. castanea | | | Q. deserticola | | |
|---|---|---|---|---|---|---|
|  | Litterfall | 30 days | 270 days | Litterfall | 30 days | 270 days |
| $Q'$ (J gMOS$^{-1}$) | 11,122 | 11,103 | 11,848 | 9,085 | 9,361 | 11,060 |
| $T_{50}$ (°C) | 396 | 379 | 375 | 400 | 377 | 375 |
| $Q_1$ (200–375 °C) (%) | 40.3 | 47.7 | 49.8 | 39.0 | 48.9 | 49.9 |
| $Q_2$ (375–475 °C) (%) | 57.0 | 49.7 | 46.1 | 55.9 | 45.5 | 43.6 |
| $Q_3$ (475–550 °C) (%) | 2.7 | 2.5 | 4.2 | 5.2 | 5.6 | 6.5 |
| $T_1$ (°C) | 347 | 344 | 343 | 352 | 342 | 350 |
| $T_2$ (°C) | 424/449 | 410 | 425 | 461 | 414 | 432 |
| $T_3$ (°C) | – | – | – | – | 490 | 502 |

**Notes.**

$Q'$, Energy released per gram of organic matter; $T_{50}$, Temperature in which the 50% of energy release occurs; Percent of energy released in: labile region $Q_1$, recalcitrant region ($Q_2$) and extra-recalcitrant region ($Q_3$). Temperature in which the peak occurs in: the labile region ($T_1$), recalcitrant region ($T_2$) and extra-recalcitrant region ($T_3$).

**Table 3** Means of thermal parameters obtained through DSC at 30 days of decomposition.

|  | Litter quality | | | | Site | | | |
|---|---|---|---|---|---|---|---|---|
|  | QcL | QxL | QdL | F(p) | QcS | QxS | QdS | F(p) |
| $Q'$ (W g$^{-1}$) | 10,815 | 10,804 | 11,083 | 0.53 (0.60) | 10,716 | 11,154 | 10,832 | 1.35 (0.32) |
| $T_{50}$ (°C) | 376 | 375 | 377 | 0.60 (0.58) | 378 | 375 | 375 | 3.99 (0.07) |
| $Q_1$ (%) | 49.2 | 49.7 | 48.8 | 0.80 (0.49) | 48.4 | 49.7 | 49.6 | 3.36 (0.10) |
| $Q_2$ (%) | 48.6[A] | 46.8[AB] | 45.6[B] | **17.8 (<0.01)** | 47.2 | 46.8 | 46.9 | 0.04 (0.95) |
| $Q_3$ (%) | 2.2[B] | 3.4[B] | 5.6[A] | **20.4 (<0.01)** | 4.3 | 3.5 | 3.4 | 0.25 (0.78) |

**Notes.**

Different capital letters indicate significant differences ($P < 0.05$) according to a one-way ANOVA model. Bold letters are statistically significant. Qc, Quercus castanea litter; Qx, Mixed species litter; Qd, Quercus deserticola litter. The suffixes -L and -S refer to litter and site, respectively. $Q'$, Energy released per gram of organic matter; $T_{50}$, Temperature in which the 50% of energy release occurs; Percent of energy released in: the labile region ($Q_1$), recalcitrant region ($Q_2$) and extra-recalcitrant region ($Q_3$). F, value of the test statistic; $p$, significance level.

litter at 270 days increased the energy released, compared to 30 days, in the Q1 region (Qc + 4% and Qd + 2%), the Q2 region showed a decrease in the litter of both species (Qc − 7% and Qd − 4%; Table 2) and the Q3 region increased for both species but more for *Q. castanea* (Qc + 63% and Qd + 15%; Table 2). The values of T50 were displaced to lower temperatures (Table 2) in both species at 270 days of decomposition in comparison to 30 days of decomposition.

In the one-way ANOVA of the thermal parameters at 30 days of decomposition, recalcitrant and extra-recalcitrant compounds were the highest and the lowest in QcL and QdL, respectively, without a significant effect of site (Table 3). Figure 4 shows the thermograms of litter at 30 days of decomposition of each *Quercus* species condition within each site. The QdL showed the highest peak in the labile region at the three sites, while the QcL showed the highest peak in the recalcitrant region also in the three sites. However, only the QdL showed a small peak in the extra-recalcitrant region, mainly at the Qd site.

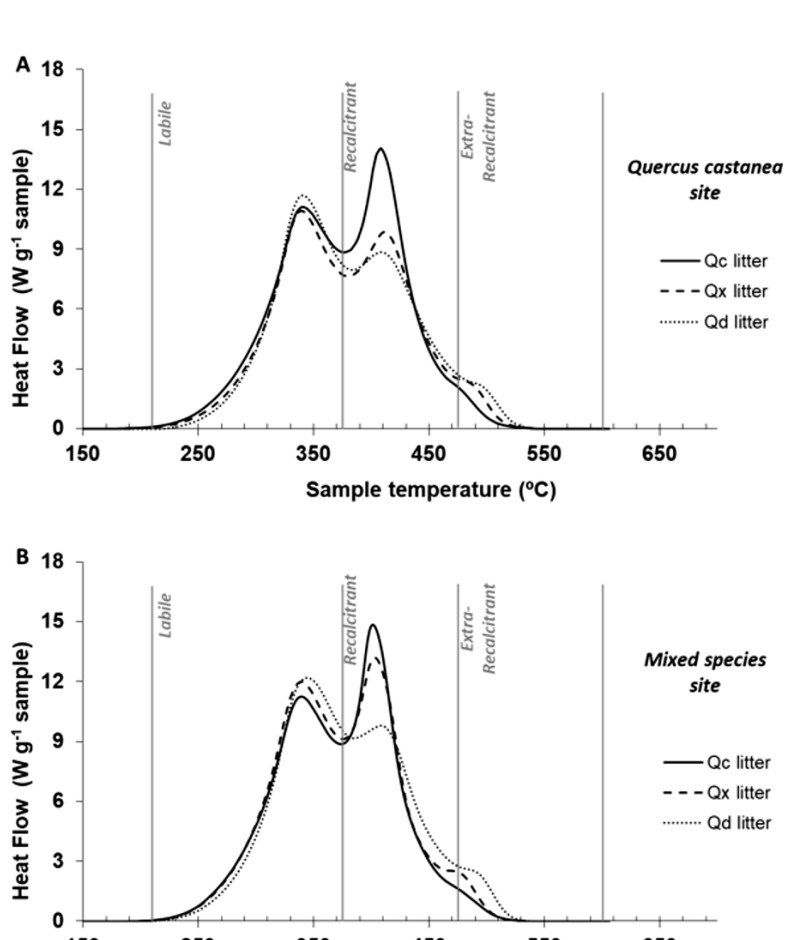

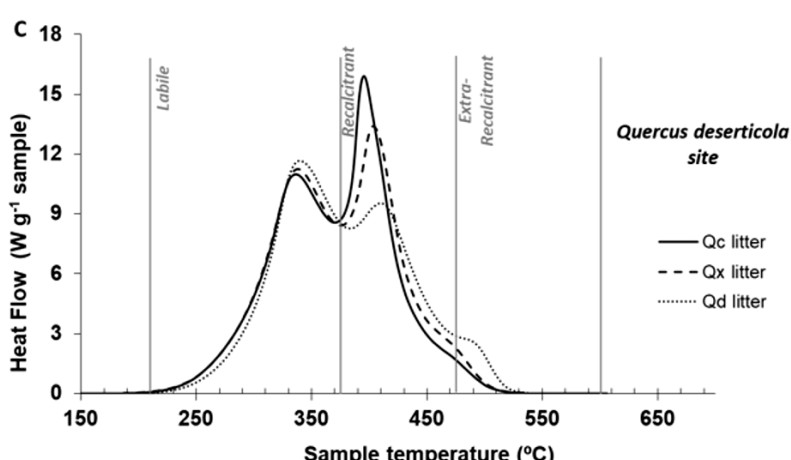

**Figure 4** **DSC thermogams of each litter condition decomposed.** DSC thermograms of each litter condition decomposed on (A) *Quercus castanea* site, (B) mixed species site and (C) *Quercus deserticola* site. *Q. castanea* litter (Qc, solid line), mixed species litter (Qx broad dotted lines) and *Q. deserticola* litter (Qd, fine dotted lines).

**Table 4** Means of the concentration of total and dissolved C, N and P in litterfall and decomposed litter at 30 and 270 days for *Q.castanea*, mixture of species and *Q.deserticola* at their site of origin.

| | *Q. castanea* | | | Mixed species | | | *Q. deserticola* | | |
|---|---|---|---|---|---|---|---|---|---|
| | Litterfall | 30 days | 270 days | Litterfall | 30 days | 270 days | Litterfall | 30 days | 270 days |
| **Total nutrients** (mg g$^{-1}$) | | | | | | | | | |
| C | 489$^a$ | 459$^{ab}$ | 404$^b$ | 471$^a$ | 459$^{ab}$ | 422$^b$ | 469$^a$ | 441$^{ab}$ | 404$^b$ |
| N | 7.3$^{Ba}$ | 7.5$^{Ba}$ | 7.5$^{Ca}$ | 8.3$^{ABb}$ | 8.7$^{Bb}$ | 12.7$^{Aa}$ | 10.3$^{Aa}$ | 10.7$^{Aa}$ | 9.8$^{Ba}$ |
| P | 0.21$^{Aa}$ | 0.31$^{Ba}$ | 0.23$^{Ba}$ | 0.26$^{Ac}$ | 0.63$^{Aa}$ | 0.43$^{Ab}$ | 0.26$^{Ac}$ | 0.66$^{Aa}$ | 0.53$^{Ab}$ |
| C:N | 68$^{Aa}$ | 62$^{Aab}$ | 55$^{Ab}$ | 57$^{Ba}$ | 53$^{Bab}$ | 35$^{Bb}$ | 46$^{Ba}$ | 41$^{Bb}$ | 43$^{Bb}$ |
| C:P | 2,316$^{Aa}$ | 1,527$^{Ab}$ | 1,806$^{Ab}$ | 1,928$^{Ba}$ | 740$^{Bb}$ | 1,024$^{Bb}$ | 1,847$^{Ba}$ | 674$^{Bb}$ | 781$^{Bb}$ |
| N:P | 34$^a$ | 25$^c$ | 34$^b$ | 34$^a$ | 14$^c$ | 30$^b$ | 40$^a$ | 16$^c$ | 19$^b$ |
| **Dissolved organic nutrients** ($\mu$g g$^{-1}$) | | | | | | | | | |
| DOC | 2,933$^{Ca}$ | 1,194$^{Cc}$ | 2,051$^{Cb}$ | 6,530$^{Ba}$ | 2,666$^{Bab}$ | 2,309$^{Bb}$ | 8,721$^{Aa}$ | 3,703$^{Ab}$ | 3,725$^{Ab}$ |
| DON | 76$^{Cb}$ | 153$^{Aa}$ | 121$^{Ba}$ | 184$^{Ba}$ | 197$^{Aa}$ | 195$^{Aa}$ | 310$^{Aa}$ | 155$^{Ac}$ | 209$^{Ab}$ |
| DOP | 26$^{Bb}$ | 32$^{Ba}$ | 10$^{Bc}$ | 33$^{ABb}$ | 62$^{ABa}$ | 14$^{ABc}$ | 41$^{Ab}$ | 73$^{Aa}$ | 30$^{Ac}$ |
| **Dissolved inorganic nutrients** ($\mu$g g$^{-1}$) | | | | | | | | | |
| NH$_4^+$ | 13$^{Cb}$ | 1$^{Bc}$ | 53$^{Aa}$ | 32$^{Ba}$ | 14$^{Ab}$ | 17$^{Bb}$ | 54$^{Aa}$ | 18$^{Ab}$ | 15$^{Bb}$ |
| NO$_3^-$ | 2.9$^{Ba}$ | 3.0$^{Aa}$ | 4.6$^{Ca}$ | 3.7$^{Bb}$ | 4.6$^{Ab}$ | 7.2$^{Ba}$ | 9.1$^{Ab}$ | 5.3$^{Ac}$ | 15.2$^{Aa}$ |
| HPO$_4^-$ | 22$^{Ca}$ | 8$^{Bb}$ | 21$^{Ca}$ | 42$^{Ba}$ | 5$^{Bb}$ | 36$^{Ca}$ | 65$^{Aa}$ | 27$^{Ab}$ | 54$^{Aa}$ |

**Notes.**
Different uppercase letters indicate significant differences ($p < 0.05$) between litter types within the same date. Different lowercase letters indicate significant differences ($p < 0.05$) between dates within the same litter type according to a repeated measures ANOVA (RMANOVA).

### Nutrient concentration and stoichiometric ratios in litterfall and decomposed litter

The concentration of C was only affected by decomposition time ($F = 3.5$; $p < 0.001$) reducing its value at 270 days of decomposition (Table 4). In contrast, the interaction between litter quality and sampling date was significant for concentrations of N and P ($F = 5.2$, $p = 0.01$ and $F = 5.5$, $p = 0.009$, respectively). The QdL had higher N concentration in all decomposition dates in comparison with QcL and QxL (Table 4). However, the QxL and the QdL had higher P concentration than QcL at 30 and 270 days of decomposition, but these values were similar in the three species litterfall (Table 4). These results suggest that the QxL and QdL had microbial P immobilization after 30 days of decomposition. Therefore, the C:N and C:P ratios were affected by litter quality and sampling date ($F = 25$, $p = 0.001$ and $F = 8.6$, $p = 0.004$, respectively); the QcL had the highest values and the 270 days of decomposition had the lowest values for the three species conditions (Table 4). However, the N:P ratio was only affected by decomposition time ($F = 17$, $p < 0.001$), where the highest and the lowest values were in the litterfall and at 30 days of decomposition, respectively (Table 4).

In contrast, the dissolved organic carbon and nitrogen (DOC and DON, respectively) concentrations were affected by the interaction of litter quality and sampling date ($F = 18$, $p < 0.001$ and $F = 21$, $p < 0.001$, respectively). The QdL had higher DOC concentration than the QcL in the two decomposition dates (Table 4), although the QxL had a higher DOC reduction at 270 days of decomposition than the QcL (64% and 30%, respectively). In contrast, the QcL had lower DON than the QxL and QdL in the litterfall and the litter at

270 days of decomposition, but the three litters had no different DON values at 30 days of decomposition (Table 4). Additionally, the QdL showed a reduction of DON concentration only at 270 days of decomposition, while the litter of the two other species conditions had increments or no changes in relation to their litterfall (QdL and QxL, respectively). The DOP concentration was affected by both litter quality and sampling date ($F = 18$, $p = 0.002$ and $F = 26$, $p < 0.00001$, respectively); the QdL and QcL had the highest and lowest DOP concentration, respectively, and the highest and the lowest DOP values were at 30 and 270 decomposition days, respectively (Table 4).

Similarly, the dissolved inorganic forms of N ($NH_4^+$ and $NO_3^-$) were also affected by the interaction between litter quality and sampling date ($F = 66$, $p < 0.001$ and $F = 9$, $p = 0.002$, respectively). Both dissolved $NH_4$ and $NO_3$ concentrations were highest in the Qd and lowest in the Qc in the litterfall, but this pattern changed at the 270 days of the experiment for $NH_4^+$ concentration (QcL>QdL = QxL; Table 4) and for $NO_3^-$ concentration (Qd>QxL>QcL; Table 4). The dissolved inorganic P form (DiP) was affected by the main factors ($F = 66$, $p < 0.001$ and $F = 26$, $p < 0.001$ for litter quality and sampling date, respectively); the QdL and QcL had the highest and lowest DiP values, respectively, and the lowest DiP values were at 30 days of decomposition (Table 4).

## Microbial activity in decomposed litter

The concentrations of microbial immobilized C, N and P (Cmic, Nmic and Pmic, respectively) were highest in Qd, followed by Qx and lowest in Qc (Fig. 5) for litter decomposed on its original site. However, the C:N, C:P and N:P microbial ratios were not affected by any factor analyzed (Fig. S1). The specific enzymatic activity of $\beta$-glucosidase (SEA BG), polyphenol oxidase (SEA POX) and dehydrogenase (SEA DHG) showed the highest values in Qc and Qx and the lowest values in Qd for litter decomposed on its original site (Fig. 6). At 30 days of decomposition, the microbial immobilization of C responded to the main factors (litter condition and site effects). The Qc site (QcS) had lower Cmic concentration than the other two sites (Fig. 5), while the Qd litter (QdL) had five-fold higher Cmic concentration than Qc (Fig. 5). However, the litter conditions promoted only differences for Nmic and Pmic. In both cases, the QdL had the highest concentrations, followed by the QxL, and the QcL had the lowest concentration values (Fig. 5).

SEA DHG was affected by both main factors (site and litter); the QcS and litter had the highest values (Fig. 6), SEA POX values were only affected by site, showing higher values in the QcS than in the QdS (Fig. 6). Meanwhile the value of SEA BG was only influenced by litter condition, with the QdL showing the lowest values (Fig. 6).

## Relation of variables with remaining mass

The thermal parameters (heat released in the combustion from the DSC analysis) at 30 days of decomposition had relationships with remaining mass; the Q2 (375–475 °C) region was positively related ($r = 0.77$, $p = 0.025$), while the Q3 region (475–550 °C) was negatively related ($r = -0.82$, $p = 0.001$). Therefore, these parameters can be used as indicators of the intensity of the decomposition process of litter (Table 5). Also, in the multiple regression model, the remaining mass at 30 days was positively explained by the C:N ratio

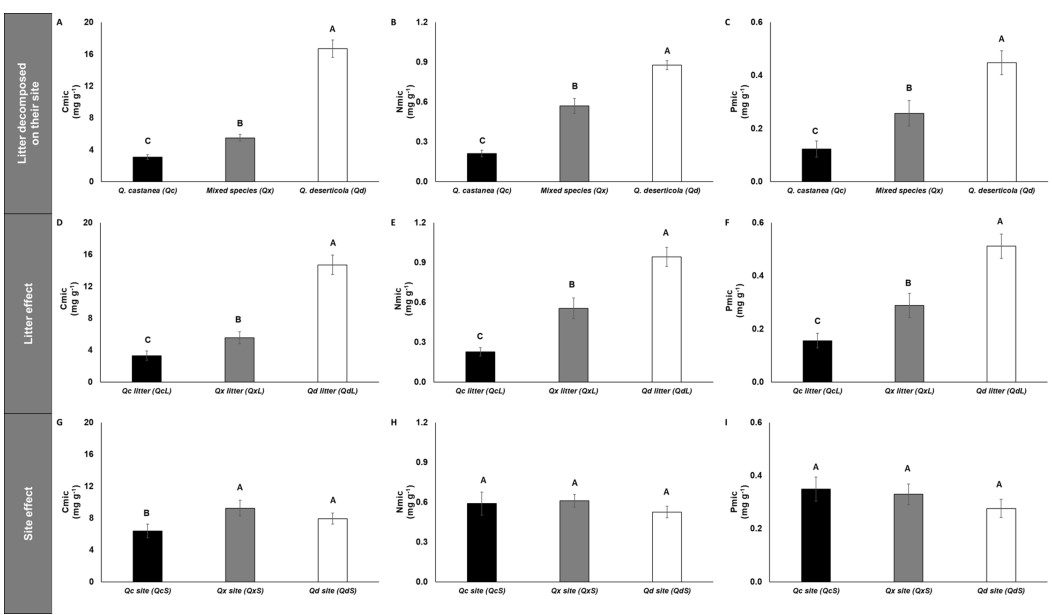

**Figure 5  Microbial immobilization of C, N and P in litter decomposed.** Microbial immobilization of C, N and P in litter decomposed on its original site (A–C), litter effect (D–F) and site effect (G–I). *Quercus castanea* (Qc), Mixed species (Qx) and *Quercus deserticola* (Qd). The suffixes -L and -S refer to litter and site, respectively. Different letters indicate statistical differences ($p < 0.05$) according to the ANOVA model.

and negatively explained by DON, DOP and dissolved $NH_4^+$ ($R^2 = 0.78$, $p < 0.001$; Table 5). In addition, the remaining mass showed a positive relation to SEA DHG and a negative relation to SEA POX ($r^2 = 0.38$, $p = 0.033$; Table 5).

## DISCUSSION

Our results indicate that the factors which regulate litter decomposition are strongly affected by the decomposition date. At the early decomposition stage (30 days) when the labile molecules, which regulate the decomposition rate (*Berg, 2014*), dominated, the coupling of litter quality and microbial activity (home-field advantage hypothesis) is the main factor. However, at the advanced decomposition stage (270 days) when recalcitrant molecules dominated, the litter decomposition is regulated by the direct effect of the chemical composition of the plant residues (substrate-matrix interaction hypothesis) and the legacy effect on the specialization of the microbial community in the use of organic compounds (functional breadth hypothesis). These conclusions are supported by the reduction of the Home-Field Advantage Index with the decomposition date (22% and 4% for 30 and 270 days, respectively). Therefore, the hypotheses that have been raised to explain the process of decomposition of the litter are not mutually exclusive (*Freschet, Aerts & Cornelissen, 2012*; *Fanin, Fromin & Bertrand, 2016*), which is only observable through cross-sowing experiments such as the one elaborated in the present study.

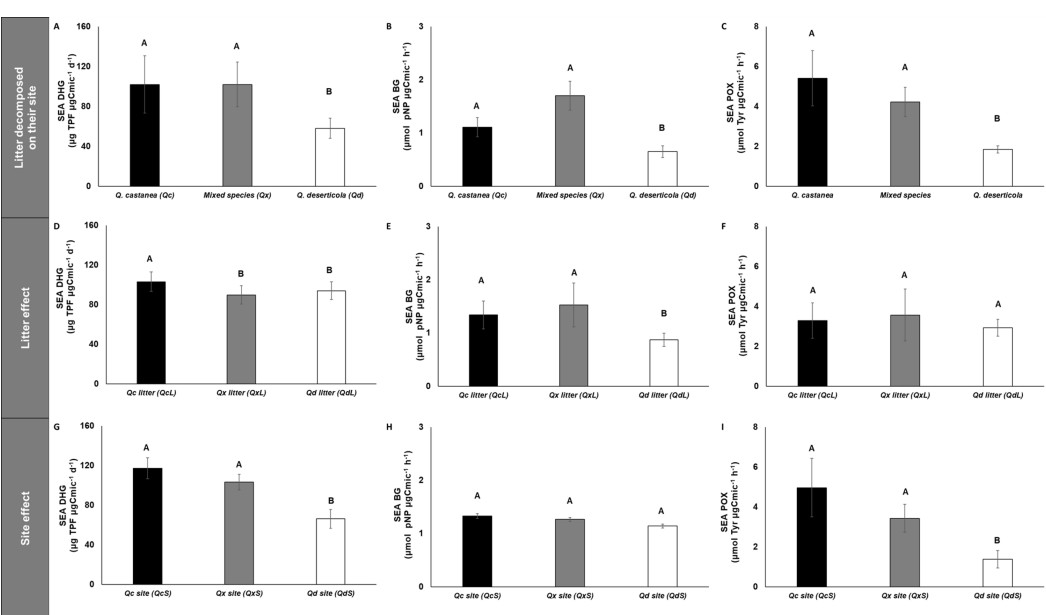

**Figure 6 Specific enzymatic activities of dehydrogenase, β-glucosidase and polyphenol oxidase in litter decomposed.** Specific enzymatic activities of dehydrogenase, β-glucosidase and polyphenol oxidase in litter decomposed on its original site (A–C), litter effect (D–F) and site effect (G–I). *Quercus castanea* (Qc), Mixed species (Qx) and *Quercus deserticola* (Qd). The suffixes -L and -S refer to litter and site, respectively. Different letters indicate statistical differences ($p < 0.05$) according to the ANOVA model.

**Table 5 Multiple regression models at 30 days of decomposition between litter remnant mass and litter chemical quality and microbial metabolism variables.**

| Factors | Included variables | Significant variables ($\beta$) | Multiple $R^2$ (p) |
|---|---|---|---|
| **Chemical quality** | C:N ratio | **0.29** | **0.78** (**<0.001**) |
| | C:P ratio | *NS* | |
| | DOC | *NS* | |
| | **DON** | **−0.56** | |
| | **DOP** | **−0.23** | |
| | **$NH_4^+$** | **−0.24** | |
| | $NO_3^-$ | *NS* | |
| | $PO_4^-$ | *NS* | |
| **Microbial metabolism** | SEA BG | *NS* | **0.38** (**0.033**) |
| | SEA CBH | *NS* | |
| | **SEA POX** | **−0.45** | |
| | SEA NAG | *NS* | |
| | SEA PHO | *NS* | |
| | **SEA DHG** | **0.41** | |

**Notes.**
Bold letters refer to significant variables according to a multiple regression model.

For example, the litter of *Q. deserticola*, when decomposed under its own canopy, showed a greater mass loss after 30 and 270 days as expected by the home-field advantage hypothesis. These results are explained by a higher concentration of total nutrient and dissolved organic forms concentrations, and lower concentration of recalcitrant compounds than in the other two sites, conditions which promotes microbial activity (*Almendros et al., 2000*; *Aponte, García & Marañón, 2013*; *Bonanomi et al., 2013*; *Fanin et al., 2013*; *Freschet et al., 2013*; *Osono, J-i & Hirose, 2013*). However, when analyzing by means of a factorial design the mass loss on the analyzed dates (30 and 270 days of decomposition), we observed that this variable is explained by the type of litter and by the site where it decomposes, since the material of *Q. deserticola* (QdL) loses more mass independently of the site where it decomposes, but it is in the sites of *Q. castanea* (QcS) and in the species mixture (QxS) where more mass is lost regardless of its origin. These results, suggest that the condition of the two other sites (QcS and QxS) decrease the importance of the home-field advantage hypothesis, mainly at 270 decomposition days.

The mechanisms that explain the higher rate of mass loss in the litterfall of *Q. deserticola* are probably related to its chemical composition, since the higher concentrations of COD and N and P in soluble forms decrease the investment in energy of the microbial community for the production of exo-enzymes to obtain organic compounds of low molecular weight (*Baldrian et al., 2010*; *Glanville et al., 2012*; *Allison, Chacon & German, 2014*).

In contrast, in places with natural incorporation of low quality material for decomposition, as the Qc site, the microbial community makes metabolic adjustments related to the chemical characteristics of the litter, which is also a key factor in the rate of decomposition. In this regard, we observed that when leaf litter of better chemical quality (QdL) is incorporated in the *Q. castanea* (QcS) site and in the mixed species site (QxS), it decomposes at a faster rate than in the site where it was produced, and also increases the immobilization of nutrients in the microbial biomass, mainly in the early decomposition stage (30 days). These results suggest that the microbial community of the litter under *Q. castanea* is more efficient in obtaining and using organic compounds, because of the continuous exposure to low quality litter (*Van Meeteren, Tietema & Westerveld, 2007*; *Baldrian et al., 2012*; *Allison, Chacon & German, 2014*; *Chavez-Vergara et al., 2014*; *Chávez-Vergara et al., 2016*), supporting the functional breadth hypothesis.

Although the litter of better chemical quality stimulated the immobilization of nutrients in the *Q. castanea* site, the microbial community of this site maintained a high SEA of POX and SEA of DG and did not modify the stoichiometric ratios in the microbial biomass. This indicates that the microbial community maintains carbon efficiency in a similar way than when it is exposed to the local litter, using more energy in the production of enzymes for the depolymerization of recalcitrant compounds than in the accumulation of biomass (*Chavez-Vergara et al., 2014*; *Chávez-Vergara et al., 2016*; *Zederer et al., 2017*). Therefore, we suggest that the greater availability of nutrients in the best quality litter (*Q. deserticola*) stimulates the growth of microbial populations, but nevertheless these populations maintain their ability to use recalcitrant compounds, in overall making the microbial community under *Q. castanea* more efficient in the decomposition of the litter, mainly at the advanced decomposition stage (270 days). This can be considered as a
legacy effect on the microbial community of low quality compounds for decomposition (*Fanin, Fromin & Bertrand, 2016*). In a previous work (*Chávez-Vergara et al., 2016*), we determined that the chemical composition of the litter influenced the composition of the fungal community. Under *Q. castanea* a greater proportion of basidiomycetes was observed, which have been reported to be specialized in the degradation of recalcitrant compounds (*Osono & Takeda, 2002*; *Snajdr et al., 2011*). Therefore, the composition of the microbial community reflects the physiological footprint of the plant (*Wickings et al., 2012*) and constitutes a legacy of the chemical traits of plant species (*Wurst, Ohgushi & Allen, 2015*; *Garcia-Palacios et al., 2016*).

In general, we can suggest that the better litter quality regulates the accessibility of organic compounds for their use (*Prescott, 2010*; *Wickings et al., 2012*; *Freschet et al., 2013*), while the microbial community, through its specialization, determines the efficiency in its use and therefore the speed of decomposition (*Strickland et al., 2009*; *Snajdr et al., 2011*; *Cleveland et al., 2014*). The above can explain the observed patterns in the thermal analysis of the decomposed material at 30 days. In the thermal analysis, we observed that it is at the Qc site that the signal of the recalcitrant compounds is more intensely decreased, and it is at this site, as mentioned above, that there is a greater investment in the production of enzymes for the degradation of recalcitrant compounds. Likewise, it is at this site that a clear signal of reactive molecules derived from microbial metabolic activity is observed, which is inferred by the appearance of an exothermic peak (ca 500 °C) in the region of extra-recalcitrant compounds (*Rovira et al., 2008*). This suggests that it is at this site that a more intense transformation of microbial organic compounds occurs, more clearly detected with thermal analysis (DSC) than in the analysis of $^{13}$C NMR.

## CONCLUSIONS

In this study, we observed that the importance of the coupling of litter quality and microbial activity depends on decomposition stage. At early decomposition stage, the home-field advantage hypothesis explained the mass loss of litter; however, in the advanced decomposition stage, the litter quality and the metabolic capacity of the microbial community can be the key drivers, mainly under *Q. castanea* conditions (litter with low available nutrients). These results enhance our knowledge about the mechanisms that regulate the decomposition of the litter in oak deciduous forests.

## ACKNOWLEDGEMENTS

The authors thank Rodrigo Velázquez, David Tolentino-Magaña and Ofelia Beltrán for field and laboratory assistance. We thank two anonymous reviewers for comments on a draft of the manuscript. B. Chávez-Vergara acknowledges the support from the Graduate Program in Biological Sciences of the National Autonomous University of México (UNAM).

### Funding

Economic support was received from project UNAM-DGAPA-PAPIIT IN206414 and IV201015 to Antonio González-Rodríguez. The funders had no role in study design, data collection and analysis, decision to publish, or preparation of the manuscript.

### Grant Disclosures

The following grant information was disclosed by the authors:
UNAM-DGAPA-PAPIIT: IN206414, IV201015.

### Competing Interests

The authors declare there are no competing interests.

### Author Contributions

- Bruno Chávez-Vergara conceived and designed the experiments, performed the experiments, analyzed the data, prepared figures and/or tables, authored or reviewed drafts of the paper, approved the final draft.
- Agustín Merino conceived and designed the experiments, performed the experiments, analyzed the data, contributed reagents/materials/analysis tools, prepared figures and/or tables, authored or reviewed drafts of the paper, approved the final draft.
- Antonio González-Rodríguez conceived and designed the experiments, performed the experiments, analyzed the data, authored or reviewed drafts of the paper, approved the final draft.
- Ken Oyama conceived and designed the experiments, contributed reagents/materials/-analysis tools, authored or reviewed drafts of the paper, approved the final draft.
- Felipe García-Oliva conceived and designed the experiments, analyzed the data, contributed reagents/materials/analysis tools, prepared figures and/or tables, authored or reviewed drafts of the paper, approved the final draft.

### Data Availability

The raw data are provided in a Supplemental File.

### Supplemental Information

Supplemental information for this article can be found online at http://dx.doi.org/10.7717/peerj.5095#supplemental-information.

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
