# Peer review of "Direct and legacy effects of plant-traits control litter decomposition in a deciduous oak forest in Mexico"

_PeerJ, doi:10.7717/peerj.5095_

## Round 0.1 · original submission · Major Revisions

Dear Felipe,

The reviewers are mostly positive about the publication of your study. However, they required a detailed revision of your manuscript. Both reviewers understand that you should work better your hypothesis and the link between these hypothesis and your findings. Please, also pay attention to your statistical test, they need to be clearly detailed in your study. See specially comments of reviewer #2.

I am looking forward a revised version of your manuscript, that has a good chance to be approved if you follow the reviewers' advice.

Reviewer 1 ·

Basic reporting

1) The English language should be improved to ensure that an international audience can clearly understand the text. Some sentences are not well structured and are difficult to comprehend.

2) In general, the introduction should highlight more the importance of this research to litter decomposition studies and ecosystem function in a micro and/or macro scale. It is not clear why your hypotheses are important to test the effects of decomposition rates in the temperate oak forest.
I suggest you to provide more details about what are the direct and legacy effects that control the decomposition process in this specific site. Also, give more examples from recent literature in the similar forest.
Specific comments:
Line 50: Fresick et al 2012 is not in the literature cited.
Lines 58 – 59: Explain the meaning of the “legacy” effect.
Lines 59 – 62: Improve this sentence with more recent literature, and give more details about the site and what these studies discovered. For example, the climate (i.e. temperature and/or precipitation) is an important factor in the regulation of litter decomposition for many ecosystems.
Lines 81 – 83: I cannot understand the sentence.
Lines 92 – 95: You should improve this sentence; I cannot understand why your study system is interesting.
Line 194: Riley & Murphy 1962 is not in the literature cited.
Line 280: StatSoft, USA is not in the literature cited.
Line 442: Baldrian et al 2012 is not in the literature cited.
Line 515: Baldrian et al 2014 is not cited in the text.

3) The structure of the article follows the author instruction, except the abstract. Please, correct your abstract as suggested in the author instruction: “Headings in structured abstracts should be bold and followed by a period. Each heading should begin a new paragraph. For example:
Background. The background section text goes here. Next line for new section.
Methods. The methods section text goes here.
Results. The results section text goes here.
Discussion. The discussion section text goes here.”

The figures and tables agree with the results.
Please, see some suggestions below:
Figure 1 A – Add 270 days in the legend of the figure.
Figure 2 – Which site was the litter residues decomposing? Close the parentheses in the last dotted line.
Figure 3 – Which site was the litter residues decomposing?
Figure 4 – Explain in the caption of the figure the meaning of Qc, Qd and Qx.
Figures 5 and 6 – Did litter decompose after 30 or 270 days of decomposition? Please, increase the resolution of the words in the axes x and y, and explain the information on axis y in the legend of the figure. One suggestion: you can use just the abbreviation in the axis x and explain them in the legend.
Table 2 – Please, add the meaning of the abbreviations in the first column and the 30 days in the legend, you wrote just 270 days.
Where is the statistical analysis of this result?
Table 3 – Please, add the meaning of the abbreviations in the first column and the second line (QcL, QxL….F(p)). Add in the legend the type of statistic test you did.
Explain in the statistic section why you did not perform this analysis in 270 days of decomposition.
Table 4 – Explain in the legend what type of statistic test you did and the different letters inside the table.
Table 5 – Why are some parameters in bold inside the table? Explain that in the legend.
Explain in the statistic section why you chose to perform this analysis in 30 days of decomposition and not in 270 days of decomposition.

I thank you for you providing the raw data, however why did you not use the results of 60 and 120 days after field exposure? You calculated the Home Field Advantage Index (HFAI) proposed by Ayres et al. (2009), this is an important result of your data because it is part of your objectives and hypotheses. It does not matter if the results are positive or negative for this index, you should explain how you calculated it and highlight the importance of this results to your study site and to the future research about this ecological theory.

4) The hypotheses B (substrate matrix interaction, SMI) and C (functional breadth hypothesis, FB) are consistent with your results, but it looks like you do not give the same importance to your first hypothesis (HFA). Consider my comments above about HFAI, and explain it in your methods (e.g. show how you calculated the HFA index based on Ayres et al. (2009) and then show the results and discussion about this parameter).

Experimental design

1) The research is consistent within the Aims and Scope of the journal.

2) You were interested to know if the litter decomposition is regulated by the direct effect (litter quality) and the legacy effect (specialization of the microbial community). Why did you not give the same importance to the home field advantage? As I mentioned above, it is not clear in your text what the relevance of your research and the knowledge gap that you want to address are. What is the advantage to test the “home field advantage theory” in your study? Finally, I suggest you to develop more arguments in your introduction based on your objectives and highlight the importance of your findings to the ecological process.

3) Your methods about chemical and nutrients parameters were described with sufficient detail and information to replicate the experiment. One suggestion is in the section 2.4 “Laboratory analysis”: add where the analyzes were conducted. However, you should provide more details about your study area. For example, the characteristics of the species in this area, the characterization of the climate (precipitation and temperature) and the soil type are important for decomposition studies, even you cited in the lines 108 and 109 that the description are in your two previous studies (Chavez-Vergara et al. 2014; Chávez-Vergara et al., 2015). Add the climate conditions of the year of your field experiment and the historical mean of annual precipitation and temperature.
You need to provide an explanation about how you performed the chemical and nutrients analyses from Quercus mixtures treatment. Did you separate the two species for these analyses? Or did you do together? What is the explanation to perform a mixed treatment?
Lastly, the experimental design is confusing, you need to improve the section 2.3 “Litterbags experiment”. See the specific comments bellow:

Line 105 – 106: Explain “oak forest fragment with low disturbance”. How big is this fragment? Are there some boarder effects in this fragment or some kind of anthropogenic pressure/activities near this area?
What do you mean by “low disturbance”? Does this area suffer some kind of management? If so, for how long? How many years ago?
Line 128 – Explain what five-select trees are. Are all five the same species and have similar structure like diameter breast height (DBH)?
Lines 126 – 132: Explain better this design. It is confusing to me about how many litterbags you harvested in 30 and 270 days after the bags were placed. How many replicates do you have for each collection?
Line 143: The exponential decay single parameter model was originally proposed by OLSEN, J. S. 1963. Energy storage and the balance of producers and decomposers in ecological systems. Ecology, 44: 322–331.

Validity of the findings

1) Statistical comments:
Provide an explanation about why you chose two different statistical analyses (lines 260 – 263), and complete the statistic section with the comments I mentioned above for tables 3 and 5 in this review.

Lines 273 – 275: Explain why you analyzed the relationship between the chemical quality of the litter and microbial metabolism variables in the remnant mass at 30 days of decomposition but not at 270 days. The labile compounds are degraded at the beginning of the decomposition, while the most recalcitrant are left to the end, so a change in the chemical composition is expected during the decomposition time (as shown in Figures 2, 3 and 4). This change could be mainly due to the specialized microbial communities that degrade the recalcitrant compounds.

Results Comments:
In the result section, do not use cited literature; use them in your discussion. Where is the result that indicates HFA?

Lines 373 and 375: Correction – NH4+ and NO3-

Discussion Comments:
All the results and discussion should answer the three hypotheses. Having that in mind, the discussion should be improved with more recent study cases found in the literature. The direct and legacy effects on decomposition of Quercus spp. should explain the research findings. For example, what results support this conclusion? You did not discuss about HFA. Since it is one of your hypothesis, you should discuss and explain it.

Lines 412 – 415: Give more details about what results show the confirmed hypotheses.
Lines 419 – 428: You will need to develop better this paragraph, I was not able to understand. The lines 419 to 423 seem to contradict the sentence bellow in lines 423 to 428. In the first sentence, “Q. deserticola showed a greater mass loss when being decomposed under individuals of this same species…than in the other two sites”, but later in the last sentence (lines 426 and 427) – “Q. deserticola (QdL) loses more mass independently of the site where it decomposes” and finally you conclude: “but it is in the sites of Q. castanea (QcS) and in the species mixture (QxS) where more mass is lost regardless of its origin.”
What species had benefits from the home field advantage, substrate-matrix interaction (SMI) and the functional breadth hypothesis? And where did the species Q. deserticola lose more mass?
Lines 437 – 439: This sentence is the opposite of the sentence above in lines 419 to 422 and lines 426 and 427.
Line 474: What site and what species?

2) This conclusion will be in accordance with your findings if you improve the discussion as I mentioned above and explain what happened with your first hypothesis about HFA.
Finally, in lines 485 and 486: why is this knowledge about the mechanisms that regulate the decomposition important in oak deciduous forests?

Reviewer 2 ·

Basic reporting

The paper explores a very interesting question, and the dataset included is impressive, using methods that have been relatively rarely used in the field of litter decomposition. This is very promising, and no additional data is needed, but the paper need a lot of rewriting before being suitable for publication.
Major comments:
1 - As it stands the links between the three tested hypotheses and the data is not explicitly presented, which makes the understanding really difficult. In the introduction, you should explicitly explain, for each hypothesis, what is expected with your dataset.
For example, something like: "If the HMI hypothesis is true, then we expect that decomposition of litter X decomposed under the cover of Y should be higher that litter X decomposed under the cover of Y". At the same time, the figures should be designed to directly show if the expectations presented in the introduction are validated.
2 - The statistics are not clear to me (see more detailed comment below) , and in particular it is unclear what you consider as your response variables and what you consider as your explanatory variables. This is particularly true for nutrients, which could be considered a response variable (nutrient loss, therefore statistically treated in the same way as mass loss) or an explanatory variables.
3 - Some of you variables (13CMNR and DSC) are perfectly relevant, but not classical in the field (in particular for DSC), and should be presented with more detail, not technically, but to explain qualitatively what kind of information it provides.
4 - I did not read all of them, but I had a look at many of the cited references, and, for some of them, the relevance seems doubtful. Please see my detailed comment below, and check content for each reference.

Detailed comments:
L 43 : I had a look at this publication Fanin et al. 2013), and found no figure explicitly linking soluble compounds to decomposition.
L 45: Are you sure that youdid not wnt to cite Talbot 2012 rather than Talbot 2014?
L 45: you directly go into a high degree of details here, with these fine data on the structure of lignin. I would begin with something more general here. See for example the recent papers by Handa 2014 Nature or Garcia Palacios 2015 Funct. Ecol.
L 41 – 48: Although you do not test it here, you should also mention the strong impact of litter physical traits (such as water holding capacity) for decomposition –see for Example Makkonen 2012 Ecol. Let.).
l 46: again, this paper explores the effect of plant-neighbor interactions on afterlife, but it is not at all the "historical paper" on this concept. See for example papers like Grime 1996.
L59: I dont understand to what extent these recent studies apparently contradict (as suggested by the term "however") the legacy effect presented in the previous sentence. At the reverse, they rather appear quite consistent the one with the other. If I am wrong, I suggest to explain why with some more details
L 82: There is a typo here, you should write “In previous studies…”
L 100: the previous sections explaining the scientific issue tackled here are clear. However, you suddenly add the mixture of litter as a new effect you want to test. This new effect hasn't been presented at all in the introduction. You should justify why you need this treatment.
L 114: It is quite unclear on how many samples you did all you analyses. You had 15 trees producing the litter. Did you characterize each of them? Did you pool them for the analyses?
L 119: It is not perfectly clear to me how you dried the litter used for the litterbag experiment?
L 122: it is interesting to have two sample dates (30 and 270 days), but you should explain why. Why did you aim at capturing short vs. longer term decomposition? This is a good idea, but needs to be justified.
L 123: Another important point that requires justification is the use of 1mm mesh size litterbag. This choice is relevant, but should be justified. By doing this, you exclude macrofauna, but not mesofauna. A brief paragraph on the actors of decomposition should be added in the introduction, leading to a justification of the choice of this mesh size.
L 129: why changing terminology here? It is not a big problem, but I don’t see the point.
L 138: Detail: the use of “remaining” seems me more common than “remnant” in this context…
L 143: Was k calculated from litter mass at t = initial, 30 days and 270 days, or with at initial and 270 days? In the latter case, with only two points, is it useful or informative to use k values? I think it is perfectly relevant use two points, but in this case, shouldn’t you rather calculate mass loss, instead of k?

L 146: although increasingly classical, I suggest to briefly introducing the main goal and objectives of 13C NMR. What does it say about litter quality? Just one sentence at the beginning of the paragraph would be helpful for a naive reader.
L 170: my previous comment is still relevant, even in a stronger way, for this paragraph on DSC. For example, this method is not presented in the classical book “Graça et al. Methods to study litter decomposition”. It does not mean at all that it is not relevant, but the use of this technic should be explained. You could add, here or in the introduction, a few lines on the “philosophy” of the technics, to what extent it might help characterizing litter quality. As it is now presented, you need to already know this not-so-classical method to really understand why you used it (for example, instead of characterizing carbohydrates, phenols, lignin, etc through classical .
Similarly, in figure 3, you should explain in the caption what is the meaning of the presented variables (Q'; Q1, etc...).
L 260:in line with my previous comment: how many repetition did you have, for example, for the characterization of the initial nutrients. Five (one for each tree)?
L 266: I don’t understand why the data on nutrients are present both line 262 and line 266. More generally, I don’t understand why data on remaining mass, nutrient concentration and dissolved nutrients are analysed in a different way than data on enzymatic activities or example. All these factors seem to have been analysed at t = 30 days and t = 270 days. In other terms, why using RMANOVA for some variables and ANOVA for others? Why including site as factor for some variables and not for others?
Moreover, these ANOVA tables are not presented in the current version. This would help in particular to understand the number of replicates.
L 273: Here, you are looking for the relationships between quality traits and a process (decomposition), which is good. However, here again, this is not clear to me, for several reasons:
1) Why did you use remaining mass at t = 30 days as a response variable, and not at t =270d?
2) Why didn’t you include all initial quality traits as explanatory variables, including variables from the 13C MNR and DSC?
I think that this part should be strongly modified or explained, but as it stands, it is really confusing. You should clearly state which your response variables are, and which your explanatory variables are.

L 284: you refer to decomposition rate in the text (which means “k”), whereas figure 1 shows remaining mass. Please clarify. If you want to present data at t = 30 and data at t = 270 separately, I suggest, again, to use mass loss as variable.

L 412 and next: the main problem of the discussion is that you validate or not some hypotheses, but since you did not explain clearly in the introduction what exactly is expected in terms of restults to validate each of you 3 hypotheses, it is very hard to understand why you state, for example, that your SMI hypothesis is validated.
l 419: wouldn't it be easier to "when decomposed under its own canopy" instead of "when beeing decomposed nder individuals of the same species"?

L 440 - l 443: you should explicitely link it with one of your 3 hypotheses.

Experimental design

As stated above, the research question is meaningful and the design seems suitable to understand the questions raised. However, the link between the question and the data is not clear enough, and should be largely improved.

The methods are described with enough technical details but information on replicate number used for each analysis is unclear.

Note that the raw data provided as supplementary material is absolutely not formatted in a suitable way. This is a working version.

Validity of the findings

Again, the results seem valid, and probably answer the question, but results should be explicitly connected to the hypotheses.
See my comments above on the raw data presentation and on statistics.

Additional comments

No additional comment

---

## Round 0.2 · accepted · Accept

I received a green flag from the reviewer. Your paper has been accepted! Congratulations!

# Reviewer 1 ·

Basic reporting

The authors considered all reviewer’s suggestion.
English was improved and the introduction highlights the importance of this research with clear hypothesis and objectives.
There is no more confusion about tables and figures legends.

Please check again if the in-text citations match the “Literature cited” section.

Specific corrections based on PDF file attached.
Line 47: The correct year is: Makkonen et al., 2012. Please correct it in the “literature cited” section as well.
Line 53 and 54: Grime and Anderson, 1986 and Grime et al. 1996 are not referenced in the “literature cited” section.
Line 54: The correct name is: Baas et al., 1989.
Line 166: The correct name is: Olson, 1963. Please correct it in the “literature cited” section as well.
Line 201: The correct year is: Angehrn-Bettinazzi et al.,1988.
Line 229: The correct citation is: Murphy & Riley, 1962.
Line 427: Correction: NH4+ and NO3-

See specific comments bellow about the “Literature cited” section:
Line 588 and 598: Baldrian and Lopez-Mondejar 2014 is not cited in text, and it is repeated in the “literature cited” section.
Line 654: The correct name is: García-Palacios.
Line 718: Talbot et al., 2014 is not cited in text.

Experimental design

The experimental design was performed and better explained, according to reviewer’s suggestions.

Validity of the findings

The results and discussion were improved and well connected to the hypotheses.